# Estimation of the force of infection and infectious period of skin sores in remote Australian communities using interval-censored data

**Michael J. Lydeamore**[1,2]*, **Patricia T. Campbell**[3,4], **David J. Price**[3,5], **Yue Wu**[6], **Adrian J. Marcato**[3], **Will Cuningham**[7], **Jonathan R. Carapetis**[6,8], **Ross M. Andrews**[7,9], **Malcolm I. McDonald**[10], **Jodie McVernon**[3,4,5], **Steven Y. C. Tong**[3,7], **James M. McCaw**[1,3,5]

1 School of Mathematics and Statistics, The University of Melbourne, Melbourne, Australia, 2 Department of Infectious Diseases, The Alfred and Central Clinical School, Monash University, Melbourne, Australia, 3 Peter Doherty Institute for Infection and Immunity, The Royal Melbourne Hospital and The University of Melbourne, Melbourne, Australia, 4 Murdoch Children's Research Institute, The Royal Children's Hospital, Melbourne, Australia, 5 Melbourne School of Population and Global Health, The University of Melbourne, Melbourne, Australia, 6 Telethon Kids Institute, University of Western Australia, Perth, Australia, 7 Menzies School of Health Research, Charles Darwin University, Darwin, Australia, 8 Perth Children's Hospital, Perth, Australia, 9 National Centre for Epidemiology & Population Health, Australian National University, Canberra, Australia, 10 Australian Institute of Tropical Health and Medicine, James Cook University, Cairns, Australia

* michael.lydeamore@monash.edu

## Abstract

Prevalence of impetigo (skin sores) remains high in remote Australian Aboriginal communities, Fiji, and other areas of socio-economic disadvantage. Skin sore infections, driven primarily in these settings by Group A *Streptococcus* (GAS) contribute substantially to the disease burden in these areas. Despite this, estimates for the force of infection, infectious period and basic reproductive ratio—all necessary for the construction of dynamic transmission models—have not been obtained. By utilising three datasets each containing longitudinal infection information on individuals, we estimate each of these epidemiologically important parameters. With an eye to future study design, we also quantify the optimal sampling intervals for obtaining information about these parameters. We verify the estimation method through a simulation estimation study, and test each dataset to ensure suitability to the estimation method. We find that the force of infection differs by population prevalence, and the infectious period is estimated to be between 12 and 20 days. We also find that optimal sampling interval depends on setting, with an optimal sampling interval between 9 and 11 days in a high prevalence setting, and 21 and 27 days for a lower prevalence setting. These estimates unlock future model-based investigations on the transmission dynamics of skin sores.

**Data Availability Statement:** The software developed to perform analysis in this manuscript is available at https://github.com/MikeLydeamore/tmi.

Data is made available in the supporting information. Estimated posterior distributions and the code used to perform simulation/estimation experiments is contained in the supporting information.

**Funding:** MJL is funded by an Australian Postgraduate Research Training Program scholarship. This work is supported by an NHMRC Project Grant titled 'Optimising intervention strategies to reduce the burden of Group A Streptococcus in Aboriginal Communities' (GNT1098319). We thank the NHMRC Centre for Research Excellence in Infectious Disease Modelling to Inform Public Health Policy (GNT1078068). JMV is supported by an NHMRC Principal Research Fellowship (GNT1117140). SYCT is supported by an NHMRC Career Development Fellowship (GNT1145033). The funders had no role in study design, data collection and analysis, decision to publish, or preparation of the manuscript.

**Competing interests:** The authors have declared that no competing interests exist.

## Author summary

Impetigo (skin sores) is a condition that remains of public health interest. Late sequelae of acute rheumatic fever and rheumatic heart disease, combined with a high prevalence in remote Australian Aboriginal communities, Fiji, and other areas of socio-economic disadvantage, mean that impetigo is a substantial contributor to the burden of disease in these settings. Despite decades of study, key quantities of interest from a transmission dynamics perspective—including the force of infection, infectious period and reproductive ratio—have not yet been determined. Such measures are arguably crucial for making informed decisions on future surveillance activities and intervention strategies. Using a series of computational and statistical methods, we find that the infectious period in remote Australian Aboriginal communities is between 12 and 20 days, and that the force of infection varies by setting. Further, we show sampling every 10 days in future surveys is optimal for further refining these estimates.

## Introduction

Infections with impetigo (commonly known as skin sores) remain highly prevalent in remote Australian Aboriginal communities, as well as Fiji and areas of socio-economic disadvantage [1, 2]. Skin sore infections in these settings are primarily caused by *Staphylococcus aureus*, and Group A *Streptococcus* (GAS). GAS is associated with post-infectious sequelae such as acute rheumatic fever and rheumatic heart disease, of which Australia has one of the highest recorded prevalences globally [1]. Despite a relatively high level of understanding about the specifics of the GAS bacterium [3–7], comparatively little is known about the natural history of skin sore infection. Furthermore, what is known is often based on historical studies from a prior generation and from a different, non-endemic, geographical region [8–11]. We aim to utilise a dynamic transmission model for skin sores to estimate two key quantities: the *force of infection*, and the *duration of infectiousness*. In the absence of information relating to immunity post-infection, we assume skin sore transmission follows the dynamics of the Susceptible-Infectious-Susceptible (SIS) model. Calculation of these two key quantities will contribute to the development and parameterisation of models which will in turn inform the design of intervention strategies aimed at reducing prevalence.

We analyse three separate datasets, all from remote Australian communities, documenting the infection dynamics of skin sores in individuals. The first dataset consists of public health network presentation data for 404 children under five years of age [12–14], collected as part of the East Arnhem Healthy Skin Project; the second contains longitudinal data for 844 individuals from three rural Australian communities, collected during household visits [15], and the third is comprised of survey visits for 163 individuals who participated in a mass treatment program [16], of which the primary endpoint was control of scabies infection. To analyse these data, we linearise the SIS model about the endemic equilibrium, and derive an expression for the likelihood of the two model parameters. By utilising Markov chain Monte Carlo (MCMC) methods, we obtain estimates of each of the force of infection, the duration of infectiousness, the basic reproductive ratio, $R_0$, and the prevalence of infection. Finally, by utilising optimal experimental design, the optimal sampling strategy to inform estimation of these parameters for use in future studies is obtained.

## Materials and methods

### Ethics statement

Ethics approval for reuse of existing data was obtained from The Human Research Ethics Committee of the Northern Territory Department of Health and Community Services and Menzies School of Health Research (Ethics approval number 2015-2516). Permission was also obtained from the custodians of each dataset. This project has been conducted in association with an Indigenous Reference Group, as well as an ongoing stakeholder group which contains Aboriginal Australian community members.

### The Susceptible-Infected-Susceptible model

We consider a stochastic representation of the Susceptible-Infectious-Susceptible (SIS) model [17]. In this model, individuals are either susceptible ($S$) or infectious ($I$). The transition rate from susceptible to infectious, known as the *force of infection*, and denoted $\lambda = \beta I/N$, where $\beta$ is the transmissibility parameter, is non-linear. This non-linearity is one of the key features of dynamic infectious disease models. However, this means that to model a population of individuals, the state of each individual is required (to know the prevalence, $i = I/N$). For the SIS model with individuals explicitly stated, the size of the required state space is $2^N$ [18]. When constructing the Markov chain representation of the SIS model then, the generator matrix, $Q$, is $2^N \times 2^N$, meaning that for large numbers of individuals, computing the matrix exponential $\exp(Qt)$, is computationally intractable. The result of this, then, is that performing inference with infectious disease models is challenging [19–24].

When the dynamics of the SIS model are at (or close to) equilibrium, then the force of infection, $\lambda$, is approximately constant. As such, we approximate the SIS model by a two-state process with a *constant* force of infection. By making this approximation, and assuming individuals are otherwise identical, it follows that a Markov chain consisting of only two states is required, independent of the underlying population size. This approximation has a straight-forward likelihood calculation, which allows estimation of parameters in a Bayesian framework and also calculation of the optimal sampling interval for future study designs through the use of optimal experimental design.

**Linearisation of the SIS model.**   The standard SIS model can be described using two transitions, infection and recovery, and two parameters, the transmissibility, $\beta$, and the rate of recovery, $\gamma$ (Table 1). Ignoring demographic processes, the total number of individuals in the population is fixed.

One of the most important quantities in infectious disease modelling is the *basic reproductive ratio*, $R_0$, defined as the mean number of secondary infection events caused by a single infectious host, in an otherwise susceptible population. The basic reproductive ratio functions as a threshold parameter, where if $R_0 \leq 1$, an outbreak of disease will not occur, while if $R_0 > 1$, then there is a non-zero probability of a disease outbreak occurring. For the SIS model, the

**Table 1. Transitions of the SIS model.** The force of infection is given by $\lambda$, the transmisibility parameter $\beta$ and the rate of recovery by $\gamma$.

| Transition | Rate |
|---|---|
| $(S, I) \rightarrow (S-1, I+1)$ | $\lambda := \beta I/N$ |
| $(S, I) \rightarrow (S+1, I-1)$ | $\gamma$ |

basic reproductive ratio is

$$R_0 = \frac{\beta}{\gamma}.$$

The quasi-equilibrium solution of the SIS model is well known [18], and the endemic prevalence of disease is

$$i^* = 1 - \frac{1}{R_0}.$$

Let $t$ be time of interest of the process. The force of infection, $\lambda(t)$ is defined as,

$$\lambda(t) := \beta i(t).$$

At equilibrium, the prevalence is approximately constant, and so the force of infection can be approximated by

$$\lambda(t) = \beta i^* \approx \lambda.$$

By performing this linearisation, it is assumed that the dynamics of disease are and remain at equilibrium. It follows that we may consider a single individual. The generator matrix for the Markov chain for the life-course of that individual is

$$Q = \begin{bmatrix} -\lambda & \lambda \\ \gamma & -\gamma \end{bmatrix},$$

and the matrix exponential of $Q$ is

$$P(t) = e^{Qt} = \frac{1}{\gamma + \lambda} \begin{bmatrix} \gamma + \lambda e^{-t(\gamma + \lambda)} & \lambda - \lambda e^{-t(\gamma + \lambda)} \\ \gamma - \gamma e^{-t(\gamma + \lambda)} & \lambda + \gamma e^{-t(\gamma + \lambda)} \end{bmatrix}. \tag{1}$$

The matrix in Eq (1), combined with an initial state and time $t$, gives the probability distribution for the Markov chain. It is possible to calculate expressions for the equilibrium prevalence, $I^*$, and the basic reproductive ratio, $R_0$, in terms of $\lambda$ and $\gamma$. Solving for the equilibrium distribution of the linearised SIS model

$$\pi Q = 0,$$

gives the equilibrium prevalence

$$i^* = \frac{\lambda}{\lambda + \gamma}. \tag{2}$$

From the standard SIS model, it is also known that the basic reproductive ratio, $R_0 = \beta/\gamma$, and $\lambda = \beta I^*$. It follows that the basic reproductive ratio, $R_0$, is given by

$$R_0 = \frac{\beta}{\gamma} = \frac{\lambda}{\gamma i^*},$$

and substituting Eq (2) gives

$$R_0 = \frac{\lambda + \gamma}{\gamma}. \tag{3}$$

Given these simple closed form expressions for the key quantities of interest, it is possible to perform estimation in a Bayesian setting, using interval-censored data.

## Data

Three separate datasets collected in Australian Aboriginal communities are considered in this study: data from public health network presentation (PHN) data on 404 children from birth to five years of age, collected as part of the East Arnhem Healthy Skin Project; data for 844 individuals from three communities, collected during household visits (referred to as the HH dataset); and data from 163 individuals who were observed for over 25 months as part of a mass treatment program in a single rural community (RC). Each dataset consists of longitudinal observations of each individual, where their infection status is recorded at each observation. The times between presentations are heavily right skewed in each dataset, with a median time to next presentation of 9 days for the PHN data, 61 days for the HH data and 119 days for the RC data. The number of observations in total is also highly variable with 13,439 observations in the PHN data, 4,507 in the HH data and 626 in the RC data. Kernel density estimates of the distribution of time until the next presentation, with the observed data overlayed, are shown in Fig 1. The suitability of each of these datasets for inferring the force of infection, λ, rate of recovery, γ, and the basic reproductive ratio, $R_0$ is investigated in Section *Verification of presentation distributions*. It is worth noting specifically that the PHN data contains information only on children from birth to five years of age, while the other two datasets contain information on individuals of all ages. Prevalence of skin sores is known to be age-dependent [25] and so by not modelling any age-structure, we are ignoring these differences.

**Data structure.**  Recall that the datasets which are considered consist of longitudinal observations for each individual, with an individual's infection status being noted as either susceptible or infected at each point. The observation is not continuous in nature, with the individual's infection status only being known at each sampling point. Data of this form are known as *interval-censored*, or *panel* data. Interval-censored data are common in epidemiology, and inference in a frequentist setting is well established [26]. Let the state of individual $i$ at observation $j$ be $X_{i,j}$, and the time at which the $j$th observation is made be $t_{i,j}$. The likelihood for a single individual, $i$, can be evaluated as

$$L_i(\lambda, \gamma) = \prod_j P_{X_{i,j}, X_{i,j+1}}(t_{i,j+1} - t_{i,j}),$$

which is the relevant entry of matrix $P$ in Eq (1), evaluated at the time difference between

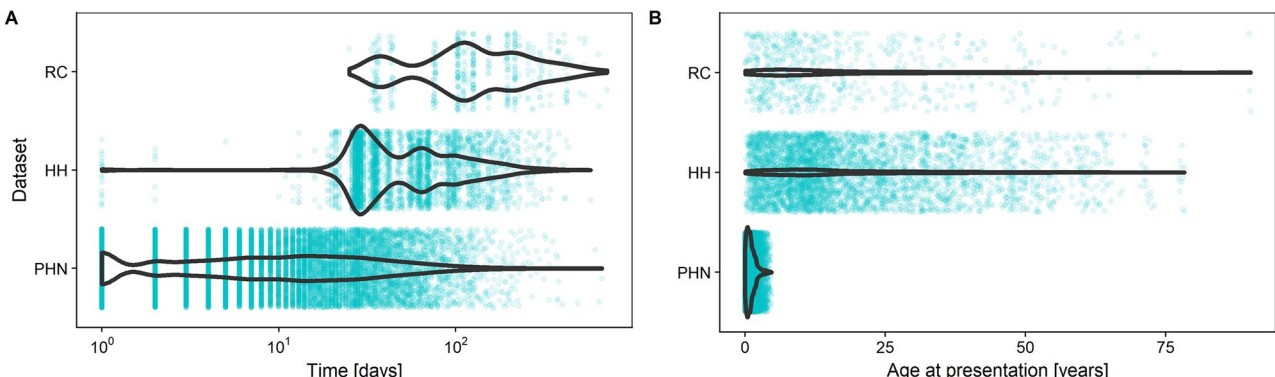

**Fig 1. Distribution of (A) time between presentations and (B) age of patients at presentation for each of the three datasets—PHN, HH and RC—with the empirical data overlayed.**

observations, $t_{i,j+1} - t_{i,j}$. It follows that the likelihood for the entire population is

$$L(\lambda, \gamma) = \prod_i L_i(\lambda, \gamma).$$

(4)

It is important to note that the likelihood in Eq (4) has assumed what is known as *ignorable* sampling times. That is, the sampling times are chosen independently of the outcome of the process. When sampling times are chosen in advance, as they were in the HH and RC datasets, then the sampling times have been proven to be ignorable [26]. For the PHN data, observations were made under what is termed a *doctor's care* scheme, whereby the next observation time is chosen at the current observation time, and based on an individual's disease state at that time. The sampling times are proven to be ignorable if the following two conditions are true [27]:

- The probability of individual $i$ being in a given disease state $u_{i,j}$ at time $t_{i,j}$, given all infection history until this point, $H_{i,j-1}$, is independent of whether an examination is carried out at this time and past examination times, and

- The conditional distribution of the $j$th observation time, $t_{i,j} = P(T_{i,j} = t_{i,j}|H_{i,j-1})$, where $T_{i,j}$ is the random variable representing the time of the $j$th infection for individual $i$, is functionally independent of the transmission parameters.

The first of these conditions effectively means that the infection status at time $t_{i,j}$ only depends on the status at time $t_{i,j-1}$ and on $t_{i,j-1}$ and $t_{i,j}$, but not on whether the individual's infection status is sampled at time $t_{i,j}$ or on previous sampling times. As treatment is prescribed by a doctor's visit, it is possible that this condition is violated. However, the dataset does not contain information on the form of treatment administered meaning that it cannot be assumed that the administered treatment is for skin sores, and almost 60% of presentations to the clinic contain no information on skin sores (and so one could assume that the primary reason for the visit is not skin sores). Further, it is noted that the estimate of infectious period in any modern setting will be reduced by the presence of treatment. As such, it is assumed that the first condition is true. The second condition means that the next observation time is conditionally independent of the transmission process. This condition is assumed to be true here due to the high frequency of presentation in this dataset, even when an individual does not have skin sores. A large number of doctors visits do not contain information about an individual's status with skin sores, including instances where an individual had been marked as infected one day prior. If we ignore these 'missing' entries, the mean time to next positive presentation following a negative presentation is 27.2 days, while the mean time to next negative presentation following a negative presentation is 23.2 days. If skin sores caused more frequent doctors presentations, then we would expect these numbers to be reversed. However, the empirical mean time to next presentation is sensitive to the frequency of missing data, and so it is unclear whether this difference can be attributed to a change in patient behaviour based on infection status, or limitations in data collection. The analysis proceeds on the basis that all sampling schemes in the given data are ignorable, but it is noted that this may not be the case.

It is important to note that in the settings studied here, treatment is routinely offered and applied to skin sores. This will augment the estimate of the recovery rate, $\gamma$, to be the average duration of infection in the presence of treatment. However, as treatment is routinely applied in many settings where skin sores are endemic, this estimate is still relevant when considering control schemes and survey designs.

Both the force of infection, $\lambda$, and the infectious period, $\gamma$ are estimated in a Bayesian context using Markov chain Monte Carlo estimation (MCMC). The MCMC is performed using

the No-U-Turn sampler implemented in Stan [28], using 10,000 iterations for 4 chains, for a total of 40, 000 iterations. The code used to perform this estimation is available at http://github.com/MikeLydeamore/TMI/. We use truncated normal distributions, $\mathcal{TN}(0.5, 0.5)$, truncated at 0, as priors for both the force of infection, $\lambda$, and the infectious period, $\gamma$. To calculate the basic reproductive number, $R_0$, we apply the formula in Eq (3) for each sample from the posterior distribution.

## Results

We start by verifying the methodology through the use of a simulation estimation study, whereby individuals are simulated from the linearised SIS model, and we attempt to recover parameters through the estimation routine detailed in the previous section. To explore the suitability of the methods and available datasets, we choose reasonable values for the force of infection, $\lambda$, and the rate of recovery, $\gamma$. In the main text, we use $\lambda = 1/60$ and $\gamma = 1/20$, while S4 Text presents results for different chosen values. After these verifications, the estimation method is applied to the observational data.

### Verification of methodology

There are multiple sources of stochastic variability in this setting, including the underlying population which is observed, the realisation from the observation distribution and the MCMC method itself. The first two of these potential causes for variation are investigated in detail here.

To investigate the variability in the underlying population, the estimation procedure is performed on 64 randomly generated populations from the linearised SIS model, and each of the 400 members of each simulated population are observed once daily for one year. This high frequency of observation means that the only source of meaningful variability is that which comes from the linearised SIS model. The top row of Fig 2 shows the marginal posterior estimates for the force of infection, $\lambda$, the rate of recovery, $\gamma$, and the basic reproductive ratio, $R_0$, for populations simulated using $\lambda = 1/60$ and $\gamma = 1/20$. Each violin plot shows an individual (marginal) posterior distribution for the parameter of interest from a randomly selected population, while the boxplot shows the variability of the posterior mean for each parameter over all 64 realised populations. The within-simulation variability is relatively high, even in this case with daily observation. However, the method estimates each parameter well and in an unbiased manner.

Next, potential variability in the observation distribution is considered. Again, a population of 400 individuals is simulated, and each simulated individual observed at times drawn from the observation distribution obtained from the PHN dataset (shown in Fig 1) over a time horizon of 1 year (Fig 2(B)). It is satisfying that although the sampling interval in the PHN dataset is notably longer than the daily case shown in Fig 2(A), the estimation method is still able to recover the simulated parameters. This suggests that oversampling the population (Fig 2(A)) gives little benefit to estimates of the parameters. Comparatively, it makes sense that if the sampling interval is too large, then no information will be gained. An example of this phenomenon is shown in Fig 3, where 20 samples are made of the population, separated by some sampling interval. The figure shows that a short sampling interval and a relatively short time horizon means that information about the parameters is difficult to recover. Similarly, a long sampling interval increases the variance in the parameter estimates. This suggests that there exists some *optimal* sampling interval. This concept will be returned to in Section *Prospective sampling strategies.*

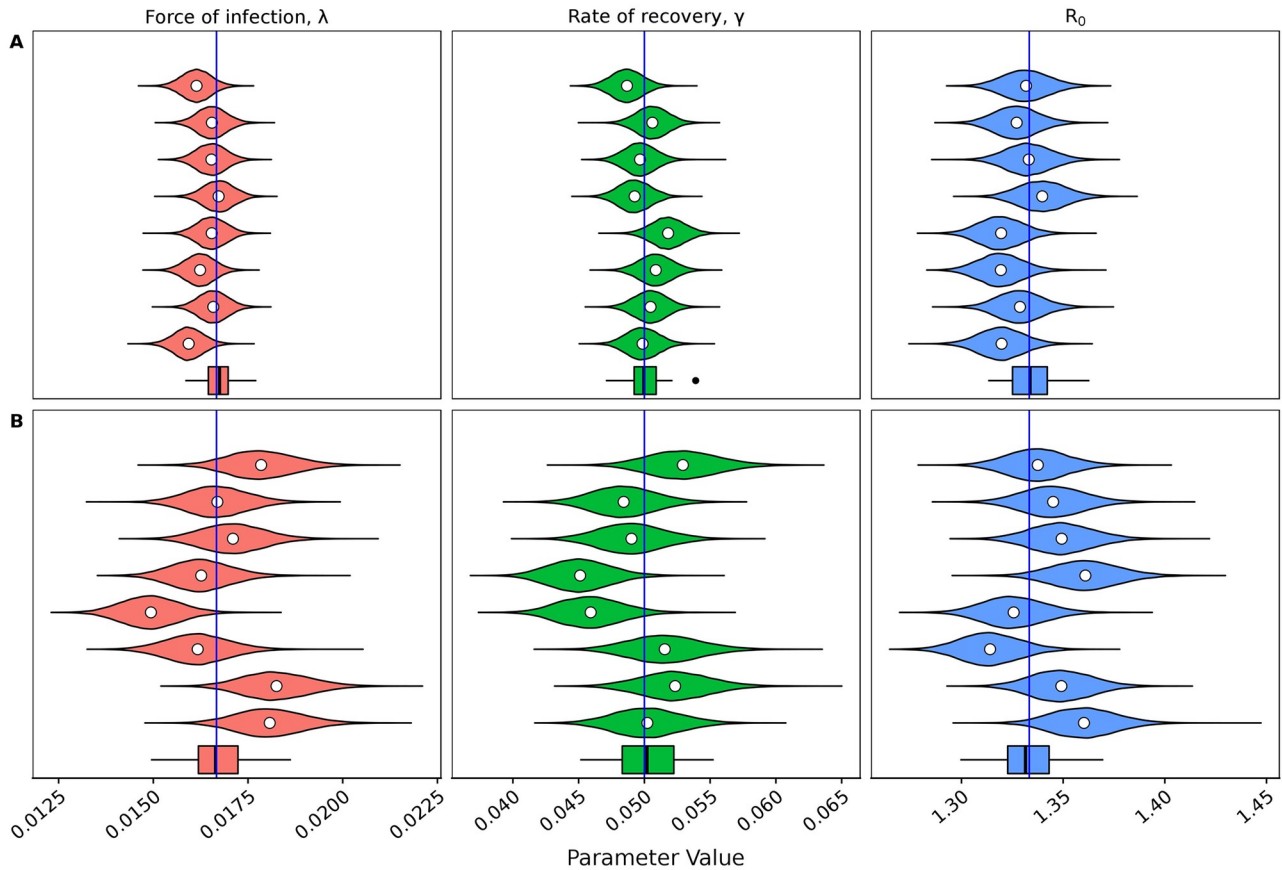

**Fig 2. Marginal posterior distributions for the force of infection, λ, the rate of recovery γ, and the basic reproductive ratio, $R_0$, from 8 randomly generated populations from the linearised SIS model under two different observation distributions.** The mean of each distribution is given by the white circle. The boxplot at the bottom of each panel represents the means of 64 marginal posteriors. The true value which was used to generate each population is represented by the blue line (λ = 1/60, γ = 1/20). The two different observation distributions are (A): Observed daily over 1 year and (B) observed according to the empirical presentation distribution from the PHN data over 1 year. Both observation distributions yield good estimates of the simulated parameters. The observation distribution from the RC dataset was tested, but has not been visualised as the estimates were far from the true values (See Fig 3).

## Verification of linearisation procedure

Having established that parameters can be re-estimated from the linearised model, we now look to verify whether the linearisation of the SIS model is valid. To do this, an individual-based implementation of the *full* (non-linearised) SIS model is used. The chosen parameters are *β* = 0.067 and *γ* = 1/20 (giving λ = 1/60 and an endemic prevalence of 25%), and 300 individuals. The Markov chain is seeded with 125 infected individuals, which is close to the equilibrium of this system. The system is run for 10 years before observation begins. The population is simulated from the full SIS model, and the estimation is performed using the *linearised* model. No stochastic extinction occurred in any of the simulations throughout this work. Fig 4 shows results from 64 realised populations, under the observation distribution from the PHN dataset. The recovery rate, *γ*, is estimated accurately and with relatively small variance. The force of infection, λ, is somewhat underestimated on average with a relative error in the mean of 15%, although the variability is large. This underestimate carries over to the estimate of the basic reproductive ratio, $R_0$. However, the true parameters are within the 95% confidence interval when averaged over the 64 simulations, similar to that seen in

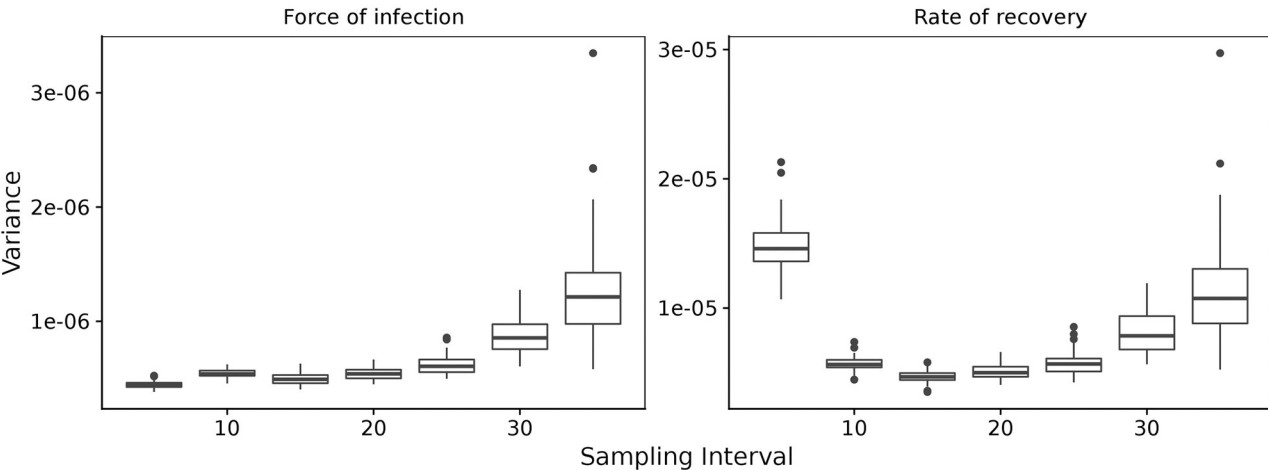

**Fig 3. Variance in the estimates of the force of infection, λ, and the rate of recovery, γ, for a range of sampling intervals.** Estimates were performed on 64 realisations of the simulated populations, each with parameters $\lambda = 1/60$ and $\gamma = 1/20$. Each realisation contains 20 observations from the simulated population, leading to the time horizon for each realisation being 20 × sampling interval.

Fig 2 under the same observation distribution. Thus, it is concluded linearisation of the SIS model is valid when the dynamics are near equilibrium.

## Verification of presentation distributions

Before estimating the force of infection, λ, and the rate of recovery, γ, for the three datasets discussed, the frequency of presentations must be checked to determine if they are sufficient for use with the method. Fig 5 shows a simulation estimation study using the presentation distributions from the PHN and HH datasets. Both datasets give good estimates. When considering the RC dataset, recall the presentation distributions shown in Fig 1. The RC dataset has a much wider sampling interval compared to the PHN and HH datasets. We suspect that this presentation distribution may not hold sufficient information to recover the parameters of interest. However, as the prevalence is observed at each survey visit, estimating the basic reproductive ratio, $R_0$, may still be possible. Fig 6 shows the prior distributions, with samples from

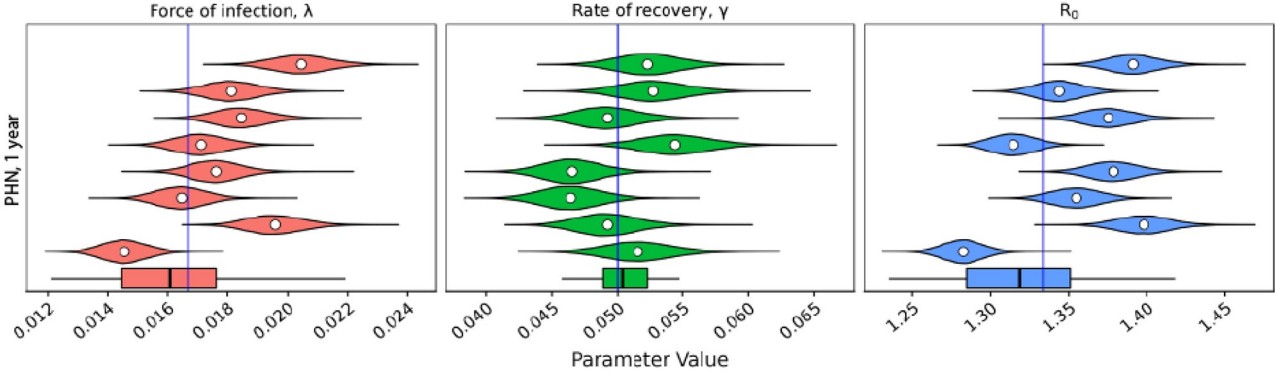

**Fig 4. Marginal posterior distributions for the force of infection, λ, the rate of recovery γ, and the basic reproductive ratio, $R_0$, from 8 randomly generated populations from the full (non-linearised) SIS model under the empirical observation distribution from the PHN data, over 1 year.** The mean of each distribution is given by the white circle. The boxplot at the bottom of each panel represents the means of 64 marginal posteriors. The true value which was used to generate each population is represented by the blue line ($\lambda = 1/60$, $\gamma = 1/20$). The simulated parameters are recovered successfully.

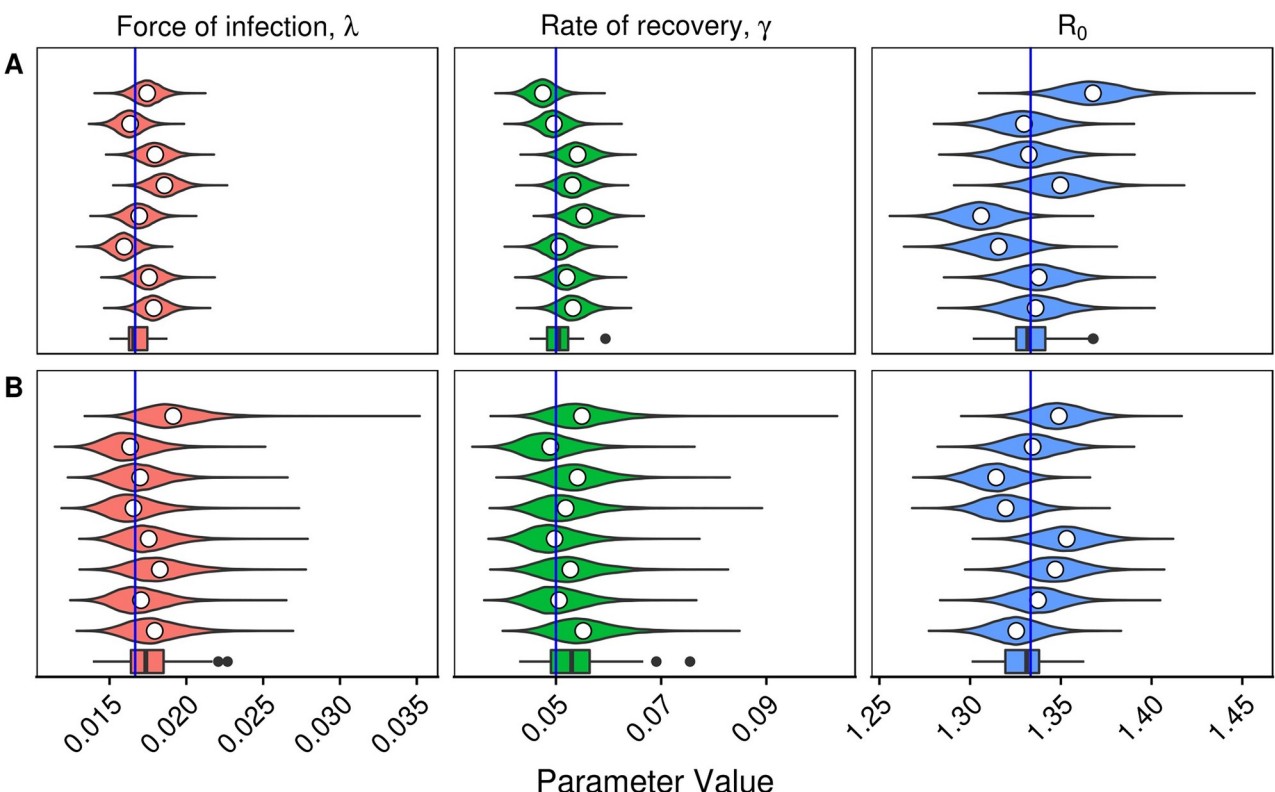

**Fig 5. Marginal posteriors for the force of infection, λ, the rate of recovery γ, and the basic reproductive ratio, $R_0$, from 8 randomly generated populations from the linearised SIS model under two different observation distributions.** The mean of each distribution is given by the white circle. The boxplot at the bottom of each panel represents the means of 64 marginal posteriors. The true value which was used to generate each population is represented by the blue line (λ = 1/60, γ = 1/20). The two different observation distributions are taken from the (A): PHN dataset (1 year) and (B) HH dataset (1 year).

the posterior distribution overlayed, under the observation distribution from the PHN dataset (panel A) and the RC dataset (panel B). Under the observation distribution from the PHN data, the posterior distribution samples are tightly clustered, with variance much smaller than in the prior distributions. Indeed, estimates are so localised relative to the prior that the samples appear to be overlayed in the figure. Comparatively, when the observation distribution is that seen in the RC dataset, the posterior samples are strongly correlated with a wide variance, indicating that this dataset does not have sufficient sampling frequency to separately estimate both the force of infection, λ, and the rate of recovery, γ. However, the posterior distribution samples align with the simulated prevalence (and thus the basic reproductive ratio, $R_0$). The RC dataset can still be used to estimate these quantities.

Having verified the suitability of each of the datasets to this estimation method, the next step is to estimate each of the force of infection, λ, the rate of recovery, γ, the prevalence of disease and the basic reproductive ratio, $R_0$.

## Estimation from data

For the PHN and HH datasets, relatively similar estimates for the infectious period, 1/γ (12 days for the PHN dataset, and 20 days for the HH dataset) are obtained. However, notably different estimates for the force of infection, λ, were obtained. In the PHN dataset, the mean

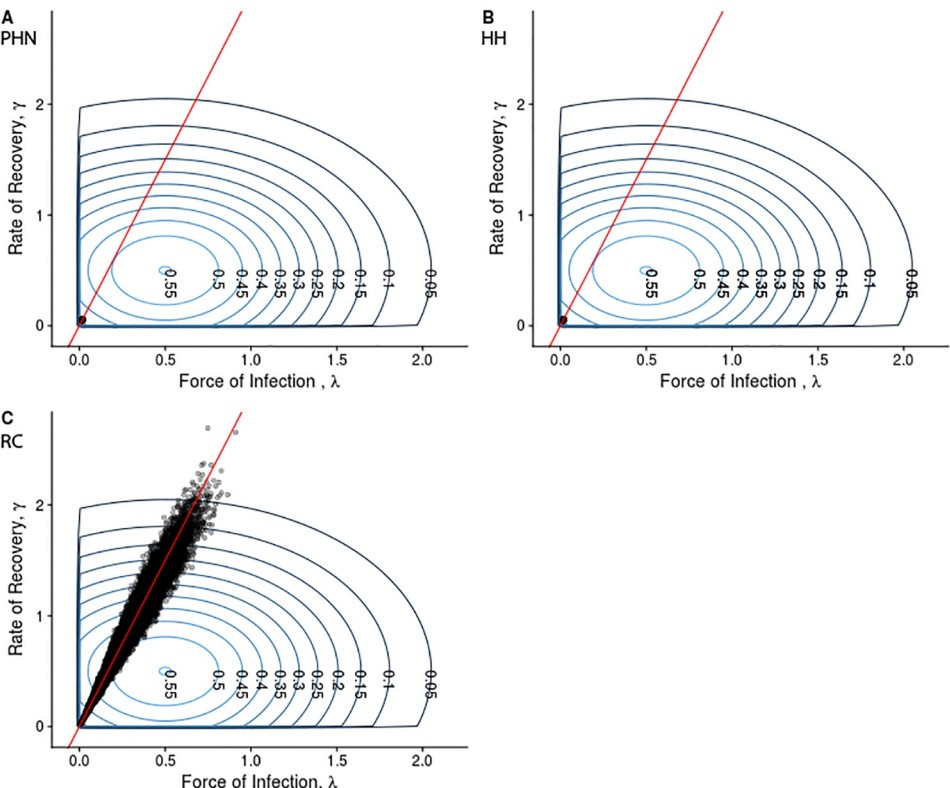

**Fig 6. Prior distribution (concentric rings) with 20,000 samples from the posterior distribution (black points) overlayed from a randomly generated population under the observation distribution from (A) the PHN dataset, (B) the HH dataset, and (C) the RC dataset.** The red line is the set of parameter values which give the true prevalence in the simulated population. In panels (A) and (B), the samples are tightly clustered with variance far smaller than the prior distribution. In panel (C), the samples are highly correlated, and with high variance, indicating the two parameters of interest cannot be uniquely determined, but their ratio (and so $R_0$) can.

force of infection is estimated at 1/20.21, while in the HH dataset, the estimate is 1/202.07—an order of magnitude different. This difference follows through to estimates of the basic reproductive ratio, $R_0$ (1.60 vs 1.10), and the prevalence, estimated to be 37.5% in the PHN dataset and only 9% in the HH dataset. For the RC dataset, $R_0$ is estimated to be 1.42, and the prevalence to be 26.9%. Point estimates of prevalence in all three study locations have been reported previously (Table 2) [15, 16, 29], at 35.6% in the region in which the PHN dataset was collected, 13.1% in the region where the HH dataset was collected and 35% in the region where the RC dataset was collected. These prevalence estimates align well with the estimates obtained using our method.

## Prospective sampling strategies

Thus far, the focus has been on previously collected datasets from which to estimate parameters. If the sole aim of a study was to collect data to *best* estimate these parameters, then the natural question to ask is when should individuals be sampled? Aided by the simple structure of the linearised SIS model, this question may be answered through optimal experimental design [30]. We take the approach of robust optimal experimental design, under the ED-optimality criterion [31, 32]. Let $\boldsymbol{\delta} = (\delta_1, \ldots, \delta_{n-1})$ define an n-sampling design with spacing $\delta_i$, $i = 1, \ldots,$

**Table 2. Parameter estimates for the force of infection, λ, and the infectious period, $1/\gamma$ from the three different datasets.** Note this method estimates the rate of recovery, $\gamma$, but the infectious period is reported here for clarity.

| Dataset | Parameter [units] | Mean | 95% CI |
|---|---|---|---|
| PHN | Force of infection (λ) [1/days] | 0.049 | (0.042, 0.059) |
| | Infectious period ($\gamma^{-1}$) [days] | 12.19 | (10.23, 14.55) |
| | $R_0$ | 1.60 | (1.56, 1.65) |
| | Prevalence | 37.5% | (31.0, 39.4) |
| Literature [29] | Prevalence | 35.6% | (32.9, 38.3) |
| HH | Force of infection (λ) [1/days] | 0.0049 | (0.0040, 0.0062) |
| | Infectious period ($\gamma^{-1}$) [days] | 19.97 | (16.19, 24.56) |
| | $R_0$ | 1.10 | (1.09, 1.11) |
| | Prevalence | 9.1% | (8.3, 10.0) |
| Literature [15] | Prevalence | 13.1% | Not provided |
| RC | $R_0$ | 1.42 | (1.34, 1.51) |
| | Force of infection (λ) | — | Not identifiable |
| | Infectious period ($\gamma^{-1}$) | — | Not identifiable |
| | Prevalence | 29.6% | (25.4, 33.8) |
| Literature [16] | Prevalence | 35% | Not provided |

$n - 1$ between subsequent observations. Then, the optimal sampling design, $\boldsymbol{\delta}^*$, is given by

$$\boldsymbol{\delta}^* = \underset{\delta}{\mathrm{argmax}} \int \det\left(I(\boldsymbol{\delta}, \boldsymbol{\theta})\right) p(\boldsymbol{\theta}) d\boldsymbol{\theta}, \tag{5}$$

where $\boldsymbol{\theta} = \{\lambda, \gamma\}$, $\boldsymbol{I}(\boldsymbol{\delta}, \boldsymbol{\theta})$ is the Fisher Information matrix, det is the determinant operator, and $p(\boldsymbol{\theta})$ is the prior distribution. Note that the optimal sampling interval, $\boldsymbol{\delta}^*$, is dependent on the prior distribution, $p(\boldsymbol{\theta})$. Two designs are considered for each dataset. The first is termed the *variable* sampling interval, where the $i$th sampling interval, $\delta_i$, is unrestricted, and $n = 11$ design spacings are chosen. Although this design strategy is optimal over a 12 visit design, adhering to the varying intervals may be difficult from an implementation perspective. A more practical strategy, and the second considered here, is termed the *fixed* sampling interval, where $\delta_i = \delta$, $\forall i$. This is equivalent to considering $n = 1$ design spacing, as the population dynamics are assumed to be at equilibrium throughout the study.

The integral in Eq (5) is approximated using a Monte Carlo estimate with 5,000 samples. Each individual is observed 12 times. We use the induced natural selection heuristic for finding optimal strategies [33]. For detail on the algorithm inputs and evidence of convergence, see S3 Text.

**Recommended sampling strategies.** We calculate the optimal strategy using the posterior distributions obtained from the PHN dataset, HH dataset and the union of the these two posterior distributions as the prior distribution in Eq (5). The results for both the variable interval strategy and the fixed interval strategy are shown in Table 3.

Under the constraint of equal observation intervals, and restricted to whole days any sampling interval between 9 days and 11 days gives a Fisher Information within 97% of the maximum for the PHN dataset. Comparatively, for the HH dataset, any sampling interval between 21 days and 28 days gives a Fisher Information within 97% of the maximum. Combining the two posterior distributions, any sampling interval between 21 and 28 days is within 97% of the maximum. However, it should be noted that a sampling interval of 23.4 days achieves only 30% of the maximum Fisher information possible in the PHN dataset, but 99% of the

**Table 3. Optimal sampling strategies (in days) using the posterior distributions obtained from the PHN and HH datasets, as well as the union of these two posterior distributions.** Two sampling strategies are considered: variable, where the time between each observation is allowed to vary, and fixed.

| Data Source | Interval | Optimal Design (days) |
|---|---|---|
| PHN | Variable | (12.9, 12.3, 10.4, 12.0, 10.3, 12.5, 12.5, 9.1, 11.3, 10.6, 11.6, 13.2) |
| | Fixed | 9.9 |
| HH | Variable | (18.1, 23.0, 30.5, 31.1, 31.6, 27.8, 30.0, 31.7, 30.0, 30.6, 32.6, 29.4) |
| | Fixed | 24.2 |
| Combined | Variable | (16.2, 23.5, 24.3, 32.0, 28.2, 26.6, 31.6, 29.2, 28.2, 28.1, 30.8, 33.9) |
| | Fixed | 23.4 |

maximum in the HH dataset. This highlights the importance of specifying the optimal sampling strategy according to the specific scenario.

Interestingly, the optimal design spacing for the fixed strategy is *not* the minimum of the optimal design spacing for the variable strategy. We propose the following hypothesis for this phenomenon: when the observation interval is allowed to vary, we can effectively 'spend' a single observation close to the previous in order to potentially gain a lot of information. However, in the fixed interval strategy, this option is not available, and so to avoid 'wasting' observations, a more conservative strategy becomes the optimal.

To understand the difference in the optimal sampling times, recall that the expression in Eq (5) maximises the Fisher Information, which through the Cramer–Rao lower bound, can be thought of as minimising the variance of the parameter estimates [34]. This estimate inherently depends on the underlying parameters of the system: when events (i.e., infection and recovery) are happening slowly (i.e., low prevalence) then sampling should happen less often, while when events are happening frequently (i.e., high prevalence), then sampling should happen more often. In the case where little prior information about the system is available, then it may be more appropriate to adopt a 'conservative' sampling strategy, which here is the faster of the two presented strategies. Doing this yields a Fisher Information of 53% of the maximum for the HH dataset. The conservative strategy is presented in S4 Text. Overall, the conservative strategy generally gives good estimation accuracy (up to 10% error in a simulation-estimation study), and so is a viable 'catch-all' strategy in the absence of prior information such as the prevalence.

## Discussion

We have provided the first model-based estimates for the duration of a skin sore infection (between 12 and 20 days), the force of infection and basic reproductive ratio (1.1 to 1.6) in three different settings. Furthermore, the optimal sampling interval for future strategies has been determined, assuming that a study's primary goal is to estimate the force of infection and duration of infectiousness. By performing the estimation in a modelling framework, the interval-censored nature of the data has been incorporated. Although the frequentist version of this estimation technique has been utilised in other disease settings [35, 36], to our knowledge this is the first time these quantities have been computed for skin sores.

Previous work on the duration of skin sore infection has estimated that under treatment, skin sore infections clear in approximately 50% of individuals in 2 days, and 85% of individuals in 7 days [37]. In this study, we do not have information on whether an individual was prescribed treatment, but it is expected that a proportion of the population were prescribed and used antibiotics. For the remainder of the population who were not treated with antibiotics, it is expected that their clearance time would be longer. As our data are a combination of treated

and untreated individuals, we propose that an exponential distribution for the duration of infectiousness is reasonable. Further, without more frequent observations, we are unlikely to be able to distinguish between other proposed distributions with confidence (S5 Text). Should accurately characterising the distribution of the duration of infectiousness be of interest, a similar approach to that in Section *Prospective sampling strategies* can be used to design a study to best discriminate between different models [38].

These results have been calculated using a linearised SIS model, in which the transmission rate has been assumed to be constant, and disease dynamics are at equilibrium. This assumption has allowed some simple analytic results which are often not able to be determined for traditional infectious disease dynamic models. However, it is important to note that the assumption of equilibrium dynamics is likely to be violated in real-world settings, particularly in the event of mass drug administration. Mass drug administration has been implemented in these communities in the past [29, 39], and was ongoing during the period of data collection in the RC dataset, although skin sores was not the primary outcome of the program in the RC setting [16]. It is also important to note that the SIS model structure, by construction, does not incorporate any period of immunity, or other potential disease states. Carriage (i.e. infected but not showing symptoms), in particular has been demonstrated for skin sore infections in the past [11, 15] and inclusion of carriage in models has been shown to substantially change predicted intervention outcomes [40]. Given the hyperendemic prevalence of skin sores in this setting, the observed high infectiousness of skin to skin transmission, and in the absence of longitudinal data related to carriage available in this study, we have ignored the carrier state in this model. Extension in this area represents an important path for future work.

It must also be noted that the condition of skin sores can be caused by a number of pathogens. The microbiology of skin sores in the Northern Territory, Australia, has been studied previously [41]. *Streptococcus pyogenes* remained the dominant pathogen, but co-infection with *Staphylococcus aureus* was present. Without microbiologic information present in our data sets, we are unable to determine which pathogen is causing infection. Accordingly, we see these results as a quantification of skin sores as a general condition.

In the populations in which these data were collected, treatment is routinely administered for skin sores. Thus, these estimates of the infectious period are inclusive of the effect of treatment, and so are likely to be lower than the *natural* infectious period (that is, in the absence of treatment being available). Although this interpretation of the infectious period is different to the *natural* infectious period, it is arguably more useful in an epidemiological context, as treatment will be given in any modern setting for a skin sores infection.

There are a number of key differences between the three datasets considered. The PHN dataset only has observations of children under five years of age. Extrapolation from this dataset to the entire population should be performed with caution as the prevalence of skin sores appears to be age-specific [25], and the average age of participants in the PHN data was younger than in the other two datasets. Despite this demographic difference, the relative similarity of the estimates of the infectious period from the HH data (in which the general population was studied) does provide some reassurance of the estimated numbers. Further, sampling times in the PHN dataset were not fixed in advance, but were rather driven by patients or health professionals. It has been assumed these sampling times are ignorable, but further investigation into this assumption may be warranted.

As well as estimation of key parameters for models of skin sores transmission, information about future experimental designs has also been provided. Although the optimal sampling interval is a function of both the force of infection and the infectious period, being able to calculate this interval provides helpful information to improve the efficiency of future study designs, or evaluation of disease control programs.

These parameter estimates unlock future model-based investigations for skin sores. By providing estimates for both the force of infection and the duration of infectiousness, more complex models which include covariates such as scabies, non-homogenous contact patterns, and population mobility can be considered, and the impact of treatment strategies in these settings can be evaluated. It is our hope that these models will lead to the development of innovative disease control measures, the application of which will reduce the burden of skin disease and health inequalities.

## Supporting information

**S1 Text. MCMC diagnostics.** MCMC diagnostics relating to convergence of the posterior distributions.
(PDF)

**S2 Text. Derivation of the Fisher Information matrix.** Derivation of the Fisher Information matrix for both variable and constant time between observations.
(PDF)

**S3 Text. Optimal sampling strategy diagnostics.** Diagnostics of the optimisation of the sampling strategies.
(PDF)

**S4 Text. Conservative sampling strategy.** Utility of the 'conservative' sampling strategy compared to the optimal sampling strategy.
(PDF)

**S5 Text. Model sensitivity.** Simulation study where the simulated data comes from a model with two infectious phases but is estimated using a single infectious phase.
(PDF)

**S1 File. Patient ages at presentation.** Patient ages at presentation, separated by dataset in which they appeared.
(CSV)

**S2 File. Posterior distributions for the three datasets.** Posterior distributions for the force of infection and infectious period from the three datasets.
(ZIP)

**S3 File. R code for performing simulation/estimation and verifying Fisher Information.** R code for performing simulation/estimation experiments, and numerically verifying the Fisher Information expressions included in the TMI package.
(ZIP)

## Acknowledgments

We would like to thank the Menzies School for Health Research and related project staff, for providing the data from the East Arnhem Healthy Skin Project, staff at the primary health care centres and the members of the remote indigenous communities for their participation. We acknowledge our partners in this work: Northern Territory Remote Health, Aboriginal Medical Services Alliance Northern Territory, Northern Territory Centre for Disease Control, One Disease and Miwatj Health and the NHMRC funded HOT NORTH initiative. We acknowledge the Lowitja Institute and the Cooperative Research Centre for Aboriginal Health who originally funded and lent significant support to the East Arnhem Healthy Skin Project.

## Author Contributions

**Conceptualization:** Michael J. Lydeamore, Patricia T. Campbell, Jodie McVernon, Steven Y. C. Tong, James M. McCaw.

**Data curation:** Yue Wu, Adrian J. Marcato, Will Cuningham, Jonathan R. Carapetis, Ross M. Andrews, Malcolm I. McDonald.

**Formal analysis:** Michael J. Lydeamore, David J. Price.

**Investigation:** Michael J. Lydeamore.

**Methodology:** Michael J. Lydeamore, David J. Price.

**Project administration:** Steven Y. C. Tong.

**Software:** Michael J. Lydeamore.

**Supervision:** Patricia T. Campbell, Jodie McVernon, James M. McCaw.

**Writing – original draft:** Michael J. Lydeamore.

**Writing – review & editing:** Michael J. Lydeamore, Patricia T. Campbell, David J. Price, Yue Wu, Adrian J. Marcato, Will Cuningham, Jonathan R. Carapetis, Ross M. Andrews, Malcolm I. McDonald, Jodie McVernon, Steven Y. C. Tong, James M. McCaw.

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
