## [Decision Letter · Decision Letter 0]

19 Sep 2019

Dear Dr Lydeamore,

Thank you very much for submitting your manuscript 'Estimation of the force of infection and infectious period of skin sores in remote Australian communities using interval-censored data' for review by PLOS Computational Biology. Your manuscript has been fully evaluated by the PLOS Computational Biology editorial team and in this case also by independent peer reviewers. The reviewers appreciated the attention to an important problem, but raised some substantial concerns about the manuscript as it currently stands. While your manuscript cannot be accepted in its present form, we are willing to consider a revised version in which the issues raised by the reviewers have been adequately addressed. We cannot, of course, promise publication at that time.

Sincerely,

Roger Dimitri Kouyos

Associate Editor

PLOS Computational Biology

Rob De Boer

Deputy Editor

PLOS Computational Biology

[LINK]

Reviewer's Responses to Questions

**Comments to the Authors:**

Reviewer #1: This article uses a very basic model to estimate important parameters for the dynamics of skin sores in remote Australian populations. I am very much in favor of these type of analysis as these parameters are essential to predict future dynamics and the effect of interventions.

However, I did not fully understand all details of the analysis and there hardly any sensistivity analysis to the model structure.

Major comments:

The analysis assumes an exponential distribution for the duration of sores. Is it possible to check this assumption? For instance, repeat the analysis with a more flexible distribution (Gamma/Weibull) and test whether the exponential distribution is supported by the data.

The authors claim that is impossible to check whether a sampling scheme is ignorable?

I do not see a reference, but I also wonder what they mean exactly with this. If having akin sores increases the rate at which individuals present to the doctor, one would expect that a statistical test whether the time between two subsequent visits is shorter in case of a negative visit followed by a positive visit compared to two negative visits. If there is a difference, this suggest that the sampling scheme is not ignorable.

In the description of the data structure, I had the impression that not all information was used. The probability P(X_{i,1}=0)=\\gamma/(\\lambda+\\gamma) and P(X_{i,1}=1)=\\lambda/(\\lambda+\\gamma). Later on, the authors discuss that the recovery rate for participants may be higher than for the general population due to treatment, but I think an earlier reference is useful.

Related to the previous point, I do not really understand how the results in Table 2 relate to the formulas on page 3: The model estimates \\lambda and \\gamma. Is R0 determined based on the prevalence at first presentation (as the recovery rate may be higher for individuals in the study due to treatment) or based on the estimates of lambda and gamma alone. If so, why are the data of the first presentation not used to obtain a better estimate of lambda and gamma

When I read that the authors linearise the SIS-model about the endemic equilibrium, I expected a different analysis than the very basic two-state Markov model they are using in which there is no dependence between individuals.

Page 6, I think the authors should mention the priors they use when they mention the MCMC procedure.

On Page 7, verification of linearization procedure? Did the observation start immediately after the seeding? Was there a conditioning on non-extinction?

More importantly, I do not really see the need for the linearization? The matrix exponent of a sparse 301x301 matrix takes 0.2 seconds on my laptop. This would mean that 40,000 iterations take a bit more than 2 hour, which is not really prohibitive.

I also do not understand the logic behind the need to verify the presentation distributions. Would a direct analysis, without any checking, not already tell you whether there is sufficient information in the data? (Based on credibility intervals for instance).

Regarding the recommended sampling strategies, is there a rule of thumb, for instance, the time between samples should be in the order of 1/\\gamma, such that the simulations do not have to be performed for each value of \\lambda and \\gamma?

Reviewer #2: # Major Comments

This is an interesting and helpful analysis of a difficult problem in infectious disease epidemiology, namely how to estimate transmission rates for endemic infections from interval-censored panel data. My primary concern with this analysis relates to the fact that asymptomatic colonization is typically assumed to be a precursor to symptomatic disease, and the SIS framework employed does not allow for colonization to impact transmission. Given that all available evidence suggests that asymptomatic individuals transmit at a rate similar to those with invasive infection. Given that this is the case, how should the estimates of R0 provided by the authors be interpreted? In particular, given that asymptomatically colonized individuals may be infectious for long periods of time, this may result in a downward bias in the estimate of the infectious period and an upward bias in the estimated force of infection at each time point. The authors should address whether this omission of colonization impacts their R0 estimates in order to ensure that their results are reliable and clinically useful.

With that said, I believe this issue is addressable via some additional assumptions about the relationship between the equilibirum prevalence of invasive disease and the equilibrium distribution of prevalence as well as the duration of colonization in the absence of invasive disease.

Other than these concerns, I do not have any other major questions about their analysis and found the section on optimal sampling to be a useful complement to the transmission modeling.

# Minor comments

1. P(t) is not defined as a probability anywhere; including this information would be helpful just from a clarity perspective.

Reviewer #3: The study entitled “Estimation of the force of infection and infectious period of skin sores in remote Australian communities using interval-censored data” by Lydeamore et al. is a first effort in quantifying key infectious diseases epidemiology parameters for impetigo, a bacterial skin infection.

The authors linearized the equations of a compartmental model to obtain simple equations for these key quantities, which were subsequently estimated in three different settings using previously collected longitudinal datasets with a Markov chain Monte Carlo procedure in a Bayesian setting.

My area of expertise being bacterial epidemiology, I am reviewing this paper from a conceptual point of view, and with the ambition to assess the real-world/biological plausibility of the authors conclusions rather than the technical aspects (statistical/computational) of their method.

Due to the multiple assumptions made by the authors (described in the following points 1,2 and 3) I am dubitative that this analysis brings a general clearer understanding of the skin sores transmission dynamics, but reporting the method, the limitations faced and results in these specific settings is a first step towards further investigation in this area.

The authors conclusions are sound and the limitations encountered generally well presented, in my opinion. The paper is neatly structured and written in a clear and accessible language.

Major concerns:

1. The observed outcome are skin sores (present/absent at the time of examination). Yet, as stated by the authors at the beginning of their introduction, both S. aureus and S. pyogenes (GAS) can cause these lesions. Within the GAS species, multiple (unrelated) serotypes are likely circulating, and there is no molecular information available to assume that one single strain is being transmitted in the communities under study. Similarly, data relative to asymptomatic colonization are absent, so the authors should be extremely careful when discussing insights into GAS transmission.

I think reformulating the last sentence of the abstract as the last sentence of the Conclusion (leaving out GAS) and clarifying this point further with a paragraph in the Conclusion (around line 343, where the immunity and carriage are discussed) is necessary. Typically, the whole picture is likely much more complex, with multiple infections by the same serotype conferring immunity to that serotype (Pandey et al. Streptococcal Immunity Is Constrained by a Lack of Immunological Memory following a Single Episode of Pyoderma, 2016, PLoS Patog.), but not at all to other strains circulating. I do not advise to advance such speculations and do not see how they could improve their model in that direction with the data available, but they should clarify the limitations of having a symptom of infection (skin sores) as outcome, rather than the actual pathogen identification.

2. Unfortunately, as stated by the authors in their conclusion, the condition of disease dynamics equilibrium on which the analysis is based is likely to be violated in real-world settings. This weakens the plausibility of the estimations obtained.

Is there any evidence in one of the three setting under observation that this condition was likely fulfilled during the period of the data collection?

3. Are the patient ages at presentation known?

If yes, why an age structure was not considered, given that impetigo is age-dependent?

In the Supporting Information section, the file “Patient ages at presentation” does not correspond to their actual age but empirical observation times. This should be corrected to avoid confusion.

In the conclusion the higher prevalence/force of infection in the young children dataset could be further discussed.

4. The choice of the parameter’s values to simulate the population requires argumentation. A table summarizing these parameters and justifying the chosen values (with citations if necessary) would be helpful for the reader.

Minor concerns:

Concerning the formulation

-Line 22, I suggest using “prior generation” instead of “generation prior” for clarity sake.

-Line 157 “It is noted that the estimate of infectious period in any modern setting will be augmented by treatment”. This sentence is unclear, no reference is given. Intuitively I would agree with the opposite: “Thus, this estimate of the infectious period is influenced by

treatment, and so is likely to be lower than the natural infectious period” (Line 351). The authors should clarify.

-Line 161 “It is important to note that it has been proven impossible to test whether or not a sampling scheme is ignorable.” This is a very strong statement, again lacking a reference to support it. The authors should reformulate this sentence and provide a reference.

-Line 219 “The results are visually similar”. This statement could be more quantitative.

-Line 230 “However, as the prevalence is observed at each survey visit,”. This sentence is unclear to me.

Concerning the data presented (figures, legends)

-Figure 1. Visualizing the population age distributions for each dataset on the side would be helpful

-Figure 2. Adding a y-axis title, such as Daily observations, 1 year/PHN empirical observation distribution, 1 year to differentiate plot A from plot B at first glance would be helpful.

Visualizing the observation distribution from the RC dataset would already highlight the point made in the following figure (even if they are far from the true values, as expected). As such, its inclusion in the figure would be welcomed.

-Figure 3. The unit [days] on the x-axis are missing. Furthermore, visualizing the corresponding time to horizon (although the calculation is as simple as Sampling interval x 20) would be helpful for the reader.

-Figure 4. Using the same x-axis values as in Figure 2 would ease comparison of the two figures.

-The legend of Figure 5 is identical to that of Figure 2, except that in legend 2, Marginal posterior distributions are mentioned while marginal posteriors are mentioned in legend 5. From my understanding, Fig 2B and Figure 5A are redundant (just posterior distributions obtained from different randomly obtained populations) is that correct? Or is there any conceptual difference between these? If yes, it is unclear. Again, labeling the y-axis would enhance the clarity of the figure at first glance.

-Figure 6. Visualizing the same plot for the HH dataset (even though it might look like the PHN one) would add some support for the reader.

**Have all data underlying the figures and results presented in the manuscript been provided?**

Reviewer #1: Yes

Reviewer #2: Yes

Reviewer #3: Yes

PLOS authors have the option to publish the peer review history of their article (what does this mean?). If published, this will include your full peer review and any attached files.

Reviewer #1: Yes: Martin Bootsma

Reviewer #2: No

Reviewer #3: No

---

## [Decision Letter · Decision Letter 1]

21 Jan 2020

Dear Dr Lydeamore,

Thank you very much for submitting your manuscript "Estimation of the force of infection and infectious period of skin sores in remote Australian communities using interval-censored data" for consideration at PLOS Computational Biology.

As with all papers reviewed by the journal, your manuscript was reviewed by members of the editorial board and by several independent reviewers. In light of the reviews (below this email), we would like to invite the resubmission of a significantly-revised version that takes into account the reviewers' comments.

We cannot make any decision about publication until we have seen the revised manuscript and your response to the reviewers' comments. Your revised manuscript is also likely to be sent to reviewers for further evaluation.

Sincerely,

Roger Dimitri Kouyos

Associate Editor

PLOS Computational Biology

Rob De Boer

Deputy Editor

PLOS Computational Biology

Reviewer's Responses to Questions

**Comments to the Authors:**

Reviewer #1: Answer 1.1. To obtain a reliable non-parametric estimate of the distribution of the duration of the infectious period, may require more data than is available. However, one could use another family of distributions and test which distribution gives the best fit (using e.g., AIC/DIC). This at least gives some hints whether the assumption of the exponentially distributed infectious period makes sense.

Answer 1.2. In [27], it is shown that based on data it can never be proven that a scheme is ignorable, i.e., there always exists a stochastic process for which the sampling scheme is ignorable. However, in this setting, the stochastic process is given (up to the values of the parameters, i.e., it is an SIS-model). Given the stochastic process, one can test whether the scheme is ignorable, so I think the stressing of the fact that one cannot determine whether a sampling scheme is ignorable is misleading in this case.

I do like the addition on the between-presentation times.

Answer 1.3/1.4

I meant that if the status at the first presentation is known, this contains also information.

If the system is in equilibrium (as is assumed), the probability that a patient is positive is \\lambda/(\\lambda+\\gamma). This information is not used in the likelihood. Later on, the authors argue that the recovery rate for participants may be higher than for the general population due to treatment, and hence, the \\gamma may change once the participant enters the cohort, but this is not discussed when the likelihood is created.

Answer 1.8: In line 52, the authors mention that the dimension of the state space is N, I think it should be N+1 (there can be 0, 1, …., N infectious individuals).

When I commented on the fact that taking the matrix exponent of an NxN-matrix is very doable when N=300, the authors replied that the dimension is actually 301^2x301^2. Why is this not corrected in the text? I also do not understand why the dimension should be 301^2x301^2. If each individual is explicitly present, I would expect 2^{N} different states and not 301^2.

New comments:

1) I noticed that the notation used in section 2 is confusing. In Table 1, S and I represent the number of individuals who are susceptible and infectious, respectively. However, the quasi-equilibrium I^* assumes that I is the fraction of the population, also the formula for R_0 assumes that the force of infection is \\beta I with I the fraction of the population who is infectious, (if I is the number of infected individuals, R0=\\beta N/\\gamma). The authors should stick to a single interpretation of I and S (either numbers or fractions) and use a different symbol (e.g., i and s) for the other.

2) The numbering of the sections is strange, there are two sections 2.1 for instance.

3) “The first of these conditions effectively means that the probability that an individual is either susceptible or infectious at time tj , given all past information, is independent of tj ,and all past examinations.” To me, this is not what the first condition means. It says that the infection status at time t_{i,j} only depends on the status at time $t_{i,j-1}$ and on $t_{i,j-1}$ and $t_{i,j}$, but not on whether is a sampling at time $t_{i,j}$ or on previous sampling times. It is, however, dependent on the time since the last known status, so it does depend on t_j.

Reviewer #2: I am comfortable with the responses to my comments and the changes the authors have made in response to them.

Reviewer #3: The authors have addressed my comments, I am satisfied with the current version of the manuscript.

**Have all data underlying the figures and results presented in the manuscript been provided?**

Reviewer #1: Yes

Reviewer #2: None

Reviewer #3: Yes

PLOS authors have the option to publish the peer review history of their article (what does this mean?). If published, this will include your full peer review and any attached files.

Reviewer #1: Yes: Martin Bootsma

Reviewer #2: No

Reviewer #3: No
---

## [Decision Letter · Decision Letter 2]

1 Apr 2020

Dear Dr Lydeamore,

We are pleased to inform you that your manuscript 'Estimation of the force of infection and infectious period of skin sores in remote Australian communities using interval-censored data' has been provisionally accepted for publication in PLOS Computational Biology.

Best regards,

Roger Dimitri Kouyos

Associate Editor

PLOS Computational Biology

Rob De Boer

Deputy Editor

PLOS Computational Biology

Reviewer's Responses to Questions

**Comments to the Authors:**

Reviewer #1: I am happy with the proposed changes. I feel the model and its limitations are discussed in suitable detail for a reader to properly judge the results.

**Have all data underlying the figures and results presented in the manuscript been provided?**

Reviewer #1: Yes

PLOS authors have the option to publish the peer review history of their article (what does this mean?). If published, this will include your full peer review and any attached files.

Reviewer #1: Yes: Martin Bootsma

---

## [Editor Report · Acceptance letter]

28 Sep 2020

PCOMPBIOL-D-19-01219R2 

Estimation of the force of infection and infectious period of skin sores in remote Australian communities using interval-censored data

Dear Dr Lydeamore,

I am pleased to inform you that your manuscript has been formally accepted for publication in PLOS Computational Biology. Your manuscript is now with our production department and you will be notified of the publication date in due course.

With kind regards,

Matt Lyles
